



# Enhancing the usability of weather radar data for the statistical analysis of extreme precipitation events

Andreas Hänsler, Markus Weiler[1]

[1]Chair of Hydrology, University of Freiburg, 79098 Freiburg, Germany

*Correspondence to*: Andreas Hänsler (andreas.haensler@hydrology.uni-freiburg.de)

**Abstract.** Spatially explicit quantification on design storms are essential for flood risk assessment and planning. Since the limited temporal data availability from weather radar data, design storms are usually estimated on the basis of rainfall records of a few precipitation stations having a substantially long time coverage. To achieve a regional picture these station based estimates are spatially interpolated, incorporating a large source of uncertainty due to the typical low station density, in

particular for short event durations.

In this study we present a method to estimate spatially explicit design storms with a return period of up to 100 years on the basis of statistically extended weather radar precipitation estimates based on the ideas of regional frequency analyses and subsequent bias correction. Associated uncertainties are quantified using an ensemble-sampling approach and event-based bootstrapping.

With the resulting dataset, we compile spatially explicit design storms for various return periods and event durations for the federal state of Baden Württemberg, Germany. We compare our findings with two reference datasets based on interpolated station estimates. We find that the transition in the spatial patterns from short duration (15 minute) to long duration (2 days) events seems to be much more realistic in the weather radar based design storm product. However, the absolute magnitude of the design storms, although bias-corrected, is still generally lower in the weather radar product, which should be addressed in

future studies in more detail.

## 1 Introduction

In the light of flood risk preparedness preparation and climate change adaptation planning there is a rising need for reliable information on the regional to local impacts of urban and sub-urban storm flows (e.g. European Flood Directive: EC, 2007 or 'Guidelines of heavy rainfall management for the federal state of Baden Württemberg': LUBW, 2016 - in German only). This

information is usually provided based on data from hydrological and hydraulic modelling chains, which themselves need spatially homogenized information on the magnitude of design storms for various duration and frequencies as input data.

In order to be able to provide reliable information, design rainfall estimates have to be based on sufficiently long time-series of rainfall observations from climate stations at a high temporal resolution (e.g. Charras-Garrido and Lezaud, 2013). Especially for the estimates of rare events (Tr>=100a) this restricts the analyses usually to a rather limited number of precipitation stations,

hence requiring substantial spatial interpolation efforts in order to regionalize the information. A further issue when dealing



with station data with very long time-series is non-stationarity of the data which requires an adaptation of the extreme value analyses (e.g. temporally dependent location parameters of the Generalized Extreme Value distribution, Cheng et al., 2014)

Apart from station data, however, temporally and spatially homogenized and station-adjusted precipitation data from weather radar become more and more available and have been used in the analysis of design storms (e.g. Overeem et al., 2009;

Haberlandt and Berndt, 2016; Panziera et al., 2016; Pöschmann et al., 2021). The main advantage of using weather radar data is the provision of a spatially complete picture of storm events on various temporal and spatial scales, as many short-term and small scale storm events are not captured by the typical network of precipitation gauges (Lengfeld et al., 2020). Hence, design storm estimates based on weather radar data are supposed to provide a more reliable spatial picture than interpolated station data.

The biggest drawback of this approach, however, is the lack of long-term weather radar records as temporally consistent data is primarily available only for the recent two decades (e.g. Saltikoff et al., 2019), hence not suitable (or only when accepting larger uncertainties) for estimating design storms with return periods larger than 30 to 40 years.

In order to overcome short records (or ungauged sites), regional frequency analysis is often used for rainfall as well as for discharge records. Based on the so called region of influence (ROI) approach (Burn, 1990), the records of a target station are

extended by pooling data from neighbouring stations located within a target-station specific region. While numerous applications of regional frequency analysis are reported for station data (e.g. Gaál and Kyselý, 2009; Requena et al., 2019), fewer examples are available for the extension of times series from weather radar. Goudenhoofdt et al. (2017) based a regional frequency analysis over Belgium on pooled radar data time-series with a sampling scheme considering radar cells in a radius of 10 km around the target cell for the extension of the precipitation records. While in general the approach lead to promising

results, the radial sampling scheme, however, lead to some artificial circular pattern in the final product and does account for the idea of similar regions based on distance alone.

A slightly different approach to conduct a regional frequency analysis is the spatial bootstrapping method (e.g. Uboldi et al., 2014). For a specific station/cell a large number of samples are established by the repeated sampling of independent events from surrounding stations/cells. This approach was recently applied to 11 years of radar data (spatial resolution of 4 km x 4

km) over the state of Louisiana, US (Eldardiry and Habib, 2020). Also in this study, the cell specific ROI, out of which the samples were pooled, was defined by the distance to the target cell. For each cell they set up 500 samples with a sample size of 11 events (in order to equal the actual number of years), each. They found that the method can provide a robust representation of extreme precipitation which is less affected by single outlier events than a non-regional pixel based approach. However, when compared with station based data, the re-sampled weather radar data has a tendency to underestimate the station records.

Reasons for this could be on one hand that the definition of the target cell specific ROI based on the distance only might not sufficient, but other factors (e.g. elevation, climate) as it is usually done with station data (e.g. Uboldi et al., 2014) should be incorporated as well. Also the fact that each sample only considers 11 events could be a source of uncertainty.

On the other hand, a general 'bias' in the weather radar when compared with stations is visible, generally increasing with rainfall intensity (Schleiss et al., 2020) as the radar precipitation is an indirect product (derived from reflectivity) integrated



over a larger area. A common approach to correct for such structural biases is the so called bias correction approach (see e.g. Maraun, 2016 for a review on bias correction) developed in climate impact research, but also already applied to weather radar data (Rabiei and Haberlandt, 2015). The basic idea behind bias correction is that structural biases in the data are removed while the specific characteristics (either spatial or temporal) are kept.

A combination of regional frequency analysis and bias correction could be a promising approach in order to generate a robust
radar based dataset for the spatially explicit estimation of design storm events. In our study we therefore apply a ROI based approach to extend a climatological record of 19 years of spatially and temporally homogenized weather radar data in combination with a station based bias correction. We focus our study regionally on the federal state of Baden Württemberg (BaWu), Germany as we have two station based, regionally interpolated design storm products available for this region that can be used to evaluate the newly generated regional design storm product based on weather radar data. Furthermore, BaWu
is topographically quite complex, which leads to a spatially rather inhomogeneous rainfall patterns (see Fig. 1a and Fig. 1c).

## 2 Data and Methods

### 2.1 Radar-based rainfall estimates

We base our work on the spatially and temporally homogenized climatological precipitation radar product of the German Weather Service referenced as RADKLIM (Winterrath et al., 2017) that is available as quasi gauge-adjusted five-minutes
precipitation product (RADKLIM_YW_V2017.002; Winterrath et al., 2018). This data consists of post-processed (artefact and attenuation correction) and station adjusted (but only hourly values) precipitation rates on a 1km x 1km grid for the time period from 2001 to 2019. Since we are mainly interested in short to medium range storm events that are mainly of convective nature, we only use data for the (summer) months from April to October, representing the main season for these kind of storm events (e.g. Ruiz-Villanueva et al, 2012; Haacke and Paton, 2021). Furthermore, the increased uncertainty connected to the
measurement of solid precipitation can be avoided when focussing on the summer season only. The original five-minute data is reassembled for different durations (e.g. 15, 60, 360 & 1440 minutes) via running sums and analysed separately for each event duration.

### 2.2 Station based reference data

For an independent reference we use two spatially interpolated design storm estimates based on precipitation station data. Both
datasets are frequently used by practitioners for the design and development of flood retention measures in Germany. The first one is the KOSTRA dataset (KOSTRA-DWD-2010R, Junghänel et al., 2017) which provides design rainfall estimates on a spatial scale of 5km x 5km for the whole of Germany for various return periods and event durations. The KOSTRA dataset was compiled by the German Weather Service and can be seen as the national standard with respect to design rainfall in Germany.





However, due to the limited spatial resolution of KOSTRA an additional dataset (available on 1km x 1km) has been compiled for the federal state of Baden Württemberg (subsequently referred to as BW-Stat; Steinbrich et al., 2016 - in German only). This dataset forms the basis for a coordinated effort of the state's environmental agency for the management of heavy rainfall and resulting pluvial floods in municipalities (LUBW, 2016; in German only). Like KOSTRA, this dataset is also based on station-specific design rainfall estimates which were spatially interpolated using a multi-linear regression approach. The finer

resolution of BW-Stat when compared to KOSTRA could be achieved by incorporating data from more stations (however often with shorter time series) and other precipitation networks than in KOSTRA into the analyses. In order to set up a sound data base at each of the locations a ROI based events pooling approach (similar to the one described in this paper) was used. Also BW-Stat design storms are available for different return periods and event durations. In order to allow a direct comparison with the radar based design storm estimates, the BW-Stat data was spatially re-interpolated (based on the original multi-linear

regression based interpolation process and identical station data) to the radar grid.

## 2.3 Regional sub-sampling

   We assume a storm event with a return period of 100 years to represent the upper end of our analysis. Therefore, we aim for a target length of the underlying time-series of about 100 years of rainfall data to meet the requirements for a profound extreme value analysis (EVA) with return periods up to 100 years. Given the 19 years of RADKLIM data already available, we need

to pool for each radar cell (cell of interest, COI) the data from four additional radar cells to statistically extent the RADKLIM data series to a respective length (95 years).

   Based on the ROI concept we defined for each radar cell a specific sampling area (with underlying sampling probabilities) that has to fulfil two criteria. On the one hand, the specific sampling area has to be located in close proximity (in terms of horizontal as well as vertical distance) to the COI in order to be spatially representative. On the other hand, we also want to make sure

that we sample additional rainfall events or intensities not necessarily present in the COI, so we have also to make sure that the sampling happens not too close to the COI.

   The underlying sampling probabilities of the specific sampling area are defined in a two-step procedure as depicted in Fig. 1b. First, a radial area around the COI is defined based on the distance to the radar cell of interest (Fig. 1, panel bI). Probabilities for this radial area are assigned based on a normal distribution with parameters mean and standard deviation of 9 km (cells)

and 6 km (cells), respectively. The maximum sampling radius was set to 25 km (cells). These numbers are chosen in order to reflect the typical size of a convective cell in Germany (about 40 km for hourly events, Lengfeld et al., 2019) but still keep the spatial representation of the sampling region for the COI.

   Second, spatial sampling probabilities are additionally defined by the respective altitude of the COI (Fig. 1, panel bII). Again, the probabilities are based on a normal distribution with the altitude of the COI as mean and a standard deviation of 50m. Note

that for the few radar cells in BaWu with an altitude above 1150 m (70 cells, see also Fig. 1a) the mean of the normal distribution was set to 1150m (instead of the altitude of the COI), in order to increase the number of possible sampling cells for these locations. Elevation ranges from 90 m to 1495 m in the study region (Fig. 1a).



In a final step, both spatial sampling probabilities are normalized with the respective maximum probability, added together
and again normalized by the maximum to generate the final spatial probability distribution for the sampling (Fig. 1, panel bIII).
Random sampling based on the underlying probabilities is now conducted out of all cells with a probability above a certain
threshold (p>=0.8). In order to prevent that neighbouring cells are sampled, the sampling probabilities of the cells in a radius
of 4 km (cells) of the cell are reduced below the threshold value after each sample is drawn (Fig. 1, panel bIV).
Although the sampling is based on a cell specific spatial distribution of probabilities the random character of the sampling
allows to sample different cells for a specific cell of interest in case the sampling is repeated. Hence it is possible to follow an
ensemble approach for the sampling in order to quantify the sampling uncertainty. For this purpose, the sampling is repeated
10 times. However, to minimize the effect of duplicated samples (cells) in the individual ensemble members, only the five
members with the lowest number of cell duplicates are selected.

### 2.4 Event definition and extreme value analysis

After the sampling process is completed, for each radar cell a data series of 95 (5 x 19) artificial years is available for multiple
durations and each of the five ensemble members. Each radar cell is hereby treated as an individual station. Although a time
series of 95 years was generated, it has to be kept in mind, that it is actually based on 19 years of weather radar rainfall
estimates, only. Hence, the concept of partial series (value over threshold concept) instead of annual series is applied to identify
the events for the EVA. The threshold value varies from cell to cell and is estimated to be the value that has a return period of
1 year using the approach of plotting positions $T_k$ for each element k of the partial series (with k =1 representing the maximum
event for the specific cell, duration and ensemble member within the 95 artificial years).

$$T_k = (L + 0.2/k - 0.4) \times (M/L) \qquad \text{(EQ 1)}$$

with M as the length of the time series in years (95 years in our case). L is the total number of independent events which is in
our case estimated by e (Euler's number) times the number of years equals 258 events . This approach is identical to the method
applied for the generation of the BW-Stat dataset (hence allows for direct comparison) and follows the guidelines for EVA
given by the German Association for Water, Wastewater and Waste (DWA, 2012). Temporal independence of the individual
events is ensured if the events are at least 48 hours apart, starting from the maximum event. This time spacing is applied for
all durations, although for short duration events this might be a rather conservative definition of independence.

For all events with rainfall rates equal to or above the threshold value, the Generalized Pareto distribution (GPD, see e.g. de
Zea Bermudez and Kotz, 2010 for details on the parameters of the GPD) is fitted (individually for each event duration, radar
cell and ensemble member) in order to be able to calculate precipitation rates for various return periods. The three (location,

scale and shape) parameters describing the GPD are estimated using the L-Moment parameter estimation method. The application of the GPD and the fitting process is similar to the approach used for the generation of the BW-Stat dataset and enables the direct comparison between our dataset and the BW-Stat estimates.

In order to estimate the uncertainty of the parameter estimation a bootstrapping method is applied for each duration, cell and
ensemble member to generate 1000 random samples of the events identified for the extreme value statistics. This results in a final total ensemble of 5000 parameter estimates for each cell and duration, hence allowing to explicitly assign confidence intervals to the estimated design storms. The chosen approach allows to eventually separate between the uncertainty range resulting from the different pooling (spanned in-between the five ensemble members) from the full range.

### 2.5 Bias-correction of RADKLIM Data

As introduced previously rainfall estimates from weather radar are known to frequently underestimate the magnitude of extreme rainfall events when compared to station data. This is usually triggered by the fact that radar measurements represent an integrated measurement of 1km x 1km while station data is a point measurement, but also other effects like an underestimation of high-intensity rainfall estimates using fixed Z-R relations for typical convective and stratiform events may play a role (e.g. Thorndahl et al., 2014). In order to compensate for such structural biases, we decided to match the magnitude
of 1yr design storms (which can be derived in a rather robust manner) of the BW-Stat dataset and the radar data (and hence also improve the comparability of the two datasets). To achieve this match of datasets, a quantile mapping approach (e.g. Cannon et al., 2015) was applied. This approach has the advantage that it corrects bias for the whole probability distribution (either temporal or spatial) to match the distribution of the target dataset but keeps the respective spatial or temporal pattern of the data.


For each station within the analysis region we select the closest four radar cells. QM correction is applied to the distribution of the location parameter (which can be taken as a proxy for a 1yr event) of the GPD of all stations and their corresponding cells for each duration separately. The resulting correction function is then applied to the frequency distribution of the location parameter of the full radar data set (again separately for each duration). All design storms are then calculated based on the
corrected location parameter, however the shape and scale parameters of the GPD have not been corrected in order to keep the consistency within the data.  The design storm estimates form bias-corrected weather radar based GPD parameters is referred to as RAD-BC whereas the non-bias-corrected version is named RAD.





## 3. Results

### 3.1 Sampling statistics

In the perfect case each radar cell would exactly occur five times (once as COI, four times as additionally selected cells) in each member of the final dataset. However, as we do not prescribe how often a cell is drawn, the relative contribution of a single radar cell to the data set deviates from the optimal case (Fig. 2 a). Overall about 90% of the radar cells contribute their events between 1 and 7 times to the final data set, with a maximum at 4 repetitions. Cells that occur more than 10 times are

extremely rare and can be neglected, leading to the conclusion that the final dataset is not biased towards single cells. With respect to the regional characteristics of the sampling, about 90% of the sampled cells have a distance to the COI in the range of 5km to 14km with a maximum occurrence at a distance of about 8km (Fig. 2b). The inter-ensemble variability in the sampling statistics is rather low (depicted by the small error bars).

   The spatial distribution of the effective ensemble size is depicted in Fig. 2c. Here we basically show how often for a COI the

identical cell has been sampled in all of the five ensembles. Although we select our five ensemble members in a way that the number of duplicates is minimized (in an optimal case only the COI would be duplicated), duplicates partly still occur and hence reduce the effective ensemble size. So in case only the COI is duplicated in all members, we set the effective ensemble size to five. If two cells (e.g. COI and one of the four sampled cells) is duplicated the effective ensemble size is set to four, etc. While for the most parts of BaWu the effective ensemble size is five it is slightly reduced along the main slopes of the

mountainous Black Forest (located in the west of BaWu) and Swabian Jura (located more the centre of BaWu, see also Fig. 1a for regional specification) regions. But only 13 out of about 41000 cells have 4 duplicated cells in the ensembles and no cell with all 5 ensemble members having the identical sub-selection of neighbouring cells occurs. Furthermore, it has to be kept in mind that even if cells are duplicated in different ensembles it doesn't necessarily mean that the EVA is based on the identical events, since the selection of the events is done subsequently to the cell sampling. Given the fact that the first event included

in the EVA is the event that has the maximum rain rate per duration out of all of the five contributing cells and further having the prerequisite that two events have to be at least 48 hours apart to be considered in the EVA it is very likely that (depending on the previously identified event) the same cell combination contributes different events to the final EVA in the different ensembles.

### 3.2 Bias correction

The impact of the quantile based correction of the location parameter is depicted in the form of cumulative frequency distributions (CFD) in Fig. 3. While the uncorrected radar data substantially underestimates the 1yr design storms, the bias corrected version overlaps (by purpose) almost perfectly to the station data when only the grid cells representing station points are included (upper row). Considering all of BaWu the comparison between interpolated station data and bias corrected radar data leads to slightly larger differences (bottom row) also partly resulting from the assumptions behind the spatial interpolation

of the station data. It has to be noted that both, BW-Stat and RAD-BC estimates, still show substantially lower rain rates for



the 1yr design storms than the KOSTRA reference dataset, for most parts of the distribution. The overestimation of extremes in the case of lower time steps can attributed to the lower spatial distribution of KOSTRA. Linked to this is also the substantially lower variability of KOSTRA, when compared with the other two datasets.

What should be kept in mind is the fact, that the applied bias correction is not having the same effect for longer return periods.

Correcting 1yr design storms only basically means that a certain rain amount is added to all events included in the EVA, hence, the relative contribution of the bias correction decreases for less-frequent design storms (see the differences between RAD and RAD-BC in Fig. 5).

### 3.3 Comparison of design storms

The spatial patterns of a 100yr design storm for four different selected event durations (15, 60, 360 and 1440 minutes) for the

two station based reference datasets (KOSTRA, BW-Stat) as well as the for the bias-corrected and re-sampled RADKLIM dataset (RAD-BC) are depicted in Fig. 4. Additionally, the absolute difference between BW-Stat and RAD-BC datasets is depicted. Note that the RAD-BC dataset represents the ensemble mean of the five individual sample products and that the data is spatially smoothed with a 3 by 3 cell filter to avoid single outliers.

In the KOSTRA dataset orographic induced patterns with elevated storm intensities along the Black Forest mountains and the

Swabian Jura as well as the Alpine foothills (see Fig. 1a for regional specification) in the far south east can be seen for short and long duration events. This pattern can be expected since the z-coordinate was incorporated (although with different weights for the different durations) in the interpolation of the station data (Junghänel et al., 2017): The 360 minute design storm in KOSTRA is actually interpolated from the 60 min and 24h (not shown) design storms. The Black Forest region is also characterized by high-intensity design storms in the BW-Stat dataset for both, short and long duration events. However,

especially for events with longer duration the dataset shows very dominant, high-intensity design storms in a region located between the Lake of Constance and the Black Forest, usually known to represent rather a rain shadow area due to fronts moving in from the west (see Fig. 1c).

The spatial patterns in the RAD-BC dataset differ quite substantially from the patterns of the two station based reference

datasets and also shows a distinct pattern change between short and long-duration events. While the spatial patterns of the 15 and 60 minute 100yr design storms show no relation to the orography or orographically induced rainfall patterns (but a slight north-south gradient) it changes in the case of the 1440 minute 100yr design storm events to a picture very similar to the April to October mean rainfall distribution. This can also be proofed with a spatial correlation analysis with the mean rainfall estimates from REGNIE resulting in an increase in the correlation coefficient from r=0.25 (15 minute events) to r=0.75 (1440

minute events). In the case of BC-Stat r remains below 0.6. Actually the spatial pattern of RAD-BC design storms is much more in line with what is expected from the underlying processes (e.g. pure convection triggered, small scale and short duration event versus more organized larger scale frontal systems for longer duration events (Lengfeld et al., 2019; Kaiser et al., 2021).





With respect to the absolute values, the direct comparison of BW-Stat and RAD-BC design storm intensities reveal that there are regions with substantially larger intensities in the RAD-BC dataset (e.g. especially in the far south east for the 1440 minute events) due to the difference in the spatial patterns. However, when integrated over the whole study region it becomes clear that although the radar dataset was bias-corrected for the 1yr design storm events, it still shows lower rainfall magnitudes for 20yr and 100yr design storms than the two station-based reference datasets.

In Fig. 5 we depict the CFD of the different datasets for four different durations and two (20yr and 100yr) return periods. We additionally included the non-bias-corrected (but spatially resampled) radar dataset (RAD; green line) in the figure to illustrate the effect of the initial bias correction. Additionally, the respective confidence interval for the RAD-BC dataset (see section on uncertainty estimation below) is included. Apart from the very high and low percentiles, the ensemble mean of the RAD-BC storm events is about 5 to 10mm lower than the respective rain rate of BW-Stat. Nevertheless, the uncertainty range spanned within the two station based reference datasets is quite large itself. While there are cases where the KOSTRA dataset lies within the confidence interval of the RAD-BC dataset (e.g. 100yr design storm with duration of 15 min), the difference to KOSTRA is sometimes even larger than to BW-Stat (e.g. 20yr design storm with duration of 360 min).

### 3.4 Uncertainty estimate

In order to be able to quantify the uncertainties for the newly developed RAD-BC dataset we conducted a twofold uncertainty analysis based on an ensemble based cell-sampling approach and classical bootstrapping  for the identification of parameter uncertainty. The confidence interval in Fig. 5 is defined by the 10th and 90th percentile of the large data sample generated by 1000 bootstraps runs for each of the 5 ensemble members (so basically combining both sources of uncertainty). The confidence band of the CFD spans about 5mm in the case of 20yr design storms and about 10mm in the 100yr case. The range of the five ensemble members only (without bootstrapping) is defined by the stippled line and accounts already for a large amount of the total uncertainty band demonstrating the importance of the ensemble based sampling approach.

The spatial patterns of the 10th and 90th percentile are rather similar to the patterns of the ensemble mean (see Fig. 6), and the uncertainty range of the respective rain rate is for most regions between 15 and 20% in in the case of 60 minute events and between 10 to 15% in the case of 1440 minute events, with relatively larger ranges in regions with lower values for the mean storm intensity. However, there are certain spots (e.g. the northern parts of the Black Forest in the case of a 100yr 1440minute design storm - framed with a dashed square in Fig. 6 - or various smaller regions in both examples) that have a slightly larger uncertainty range, although the mean storm intensities are large as well. In order to reveal the uncertainty contribution resulting from the ensemble sampling we highlighted regions with a large (> 75% of the range) contribution of the sampling uncertainty. Generally, the contribution of the sampling uncertainty is larger in regions with a lower overall uncertainty range. However, there are various spots with relatively larger uncertainty that are dominated by the sub-sampling uncertainty. The previously mentioned enhanced uncertainty in the northern Black Forest case seems to be substantially influenced by sampling uncertainty



in its eastern parts, whereas the uncertain parameter fitting dominates in the central and western parts, indicating a rather inhomogeneous pool of heavy rainfall events in these regions.

## 4. Discussion

The major benefit of the RAD-BC dataset is certainly the possibility to derive spatially homogenized heavy rainfall estimates also for less frequent (up to 100 yr return period) events. A comparison with the station based spatially interpolated reference

products revealed that the spatial patterns of the design storm events with various durations fit much better to the theoretically expected spatial patterns (random pattern for short duration events versus large-scale, orography oriented patterns for longer duration rain events) than the interpolated station products. However, the tendency to underestimate the magnitude of design storms especially for less frequent events in comparison to the reference datasets is something which should be examined in further detail.

One has to keep in mind, however, that a direct one to one comparison is only possible with the BW-Stat data since the KOSTRA data has on the one hand a much lower resolution and is based on a different set of rainfall stations. On the other hand, KOSTRA uses a different EVA approach based on a two parameter GEV distribution (Junghänel et al., 2017).

When comparing the non-bias-corrected (scale and shape) parameters of the GPD of BW-Stat and of a single member of RAD-

BC (Fig. 7) it can be seen that for the short durations (15 and 60 minute events) the scale parameter is lower in the RAD-BC data. For the long (1440 min) events, however, the deviations in the magnitude of the design storms seem to result mainly from the shape factor which is lower in RAD-BC. The intermediate 360 minute events are affected by both effects. This finding is also true when looking at various topographic sub-regions and other ensemble members of the RAD-BC dataset.

The lower values for the scale/shape parameters of RAD-BC can partly be attributed to the fact, that for high rainfall intensities

radar data is known to underestimate rainfall amounts due to the reflexivity bounds (e.g. Schleiss et al., 2020). This is only partly corrected for by the applied bias correction of the location parameter as it is an additive correction which corrects less frequent events relatively less than the more frequent events. On the other hand, a multiplicative correction would disrupt the homogeneity of the sampled events of the radar data. It is also questionable if a correction factor derived from the correction of 1yr events can be applied to events with a much lower frequency. Further it has to be kept in mind that the BW-Stat data

itself is an indirect product with events pooled from surrounding stations. While the location parameter still can be seen as a rather robust, it is questionable if the derived scale and shape parameters could be used with the same reliability for the bias correction of the radar data.

A promising way to proceed could be to only use a small subset of stations that have a reasonable long record to develop a frequency and duration specific correction function which could then be regionally applied to the radar data. However, for

BaWu there are only two stations with high temporal precipitation records available with a data series length of more than 50 years (Steinbrich et al., 2016) proposing a major challenge for this approach. Another possible approach would be to use a

weather radar product that is compiled at a higher spatial resolution. This could have the positive aspects that the difference between point versus area measurements become smaller. However, a real benefit would only be achieved if the deviations between in rainfall estimates of weather radar and station data are not increasing with rainfall intensities, which basically calls
for a non-static application of the Z-R relation in the weather radar product.

## 5 Conclusions

We present an ROI based approach to prolongate a 19yr climatological weather radar dataset of rainfall estimates in order to enhance its usability for the development of region specific design storm events. The established method has various positive aspects. The main improvement is the development of a spatially homogeneous dataset that allows for the calculation of rare
extreme events that is not dependent on spatial interpolation methods that is often the main error source when building a regional dataset based on station data. Moreover, the chosen sampling approach allows on the one hand to control the sampling region based on physical aspects. It further prevents that artificial structures previously reported in literature (e.g. development of circle structure, Goudenhoofdt et al., 2017) are dominant in the final dataset. Due to the combination of an ensemble-based sampling approach and a bootstrapping based parameter estimation an explicit designation of associated uncertainty ranges is
possible which is a major added value for the application by practitioners.

Nevertheless, the current version of the data still has some shortcomings that need to be addressed in future. While the applied bias correction approach substantially improved the outcome, the deviation to the two existing station based reference datasets in the case of the less frequent events is still something that has to be clarified in the near future. In order to improve the compatibility with the KOSTRA dataset it might be worthwhile to apply the KOSTRA EVA to the resampled event database
which underlies RAD-BC. Furthermore, the previously proposed training of the RAD-BC dataset on some high-quality long-term temporally highly resolved station data could be a way forward to enhance the credibility of the RAD-BC dataset.

## Author contributions

AH and MW jointly designed the experiment. All data analyses have been conducted by AH. The interpretation of the results as well as the drafting of the manuscript was conducted jointly by AH and MW.

**Acknowledgements**

This work was conducted within the research activities on heavy rainfall at the Chair of Hydrology, University of Freiburg, Funding for these research activities are provided by the State Office for the Environment, Measurements and Nature Conservation of the Federal State of Baden-Württemberg (LUBW) as well as the Regierungspräsidium (governing council) Stuttgart.

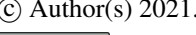



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

**Figures**

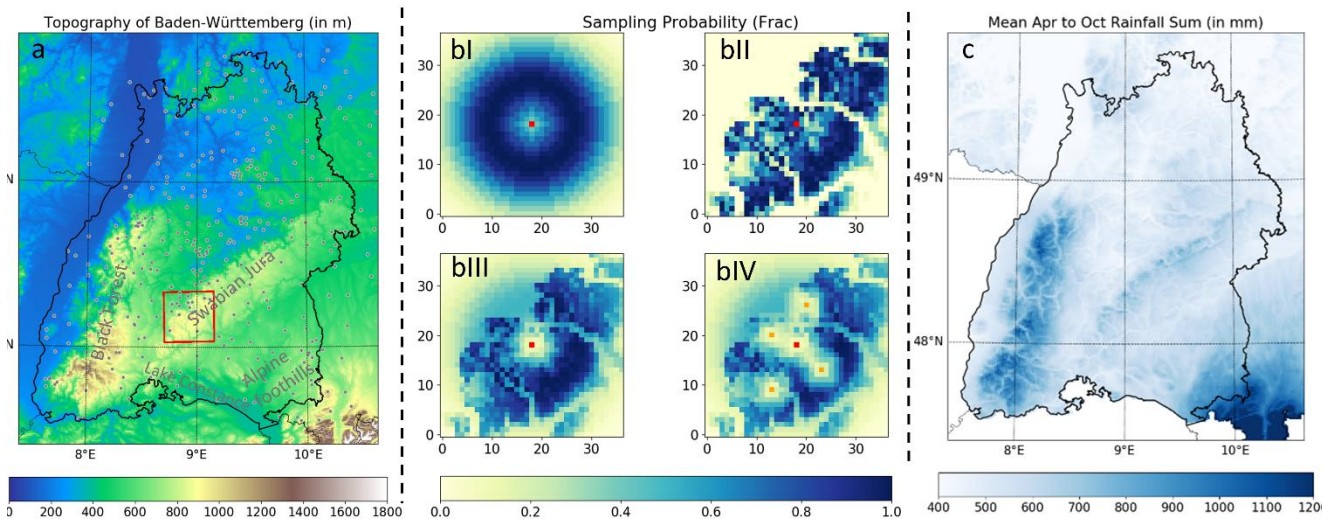


**Figure 1: (a): Topography of Baden Württemberg (BaWu) as well as location of the precipitation gauges used in the BW-Stat dataset and some of the geographical regions referred to in the text. (b): Probability for a specific radar cell to be sampled based on distance to cell of interest(bI), orography (bII) and orography and distance combined (bIII). Final sampled cells (orange) and reduced probabilities around the selected cells are depicted in panel bIV. All panels reflect the area indicated with a red square in the left**
**part of the figure. The respective cell of interest is marked with in red. (c): April to October rainfall sum of the REGNIE (Regionalisierte Niederschlagshöhen) dataset compiled by the German Weather Service.**

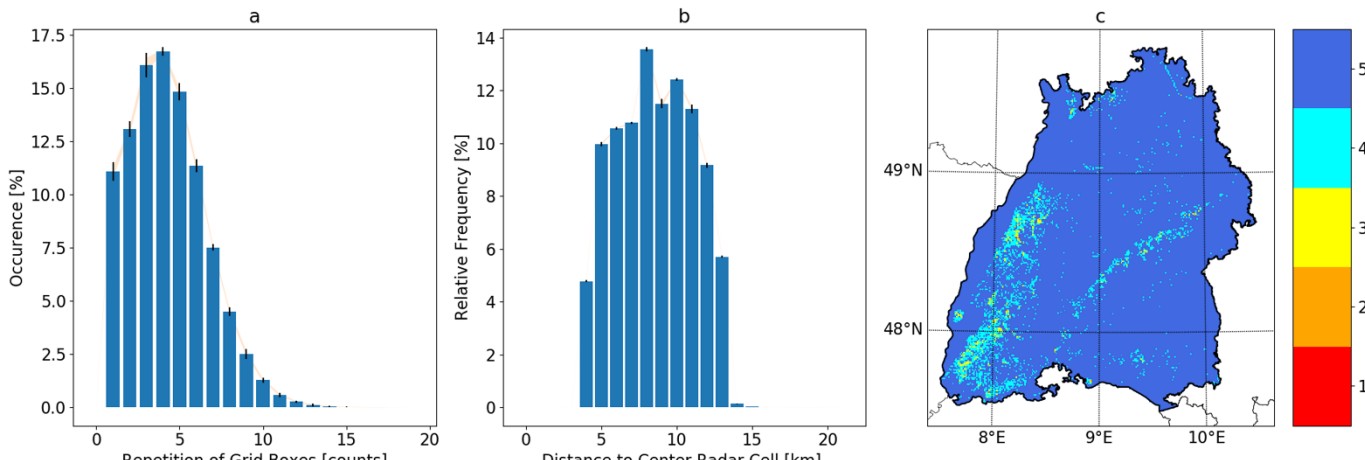





**Figure 2: Distribution of the Fraction of (a) cumulative occurrences of individual radar cells in the final sample; (b) the distance of**
**the sampled radar cells to the cell of interest. (c) Spatial distribution of the effective ensemble size. The distributions in a and b are**
**for single ensemble member, while the error bars indicate the rather small variation among the five ensemble members.**

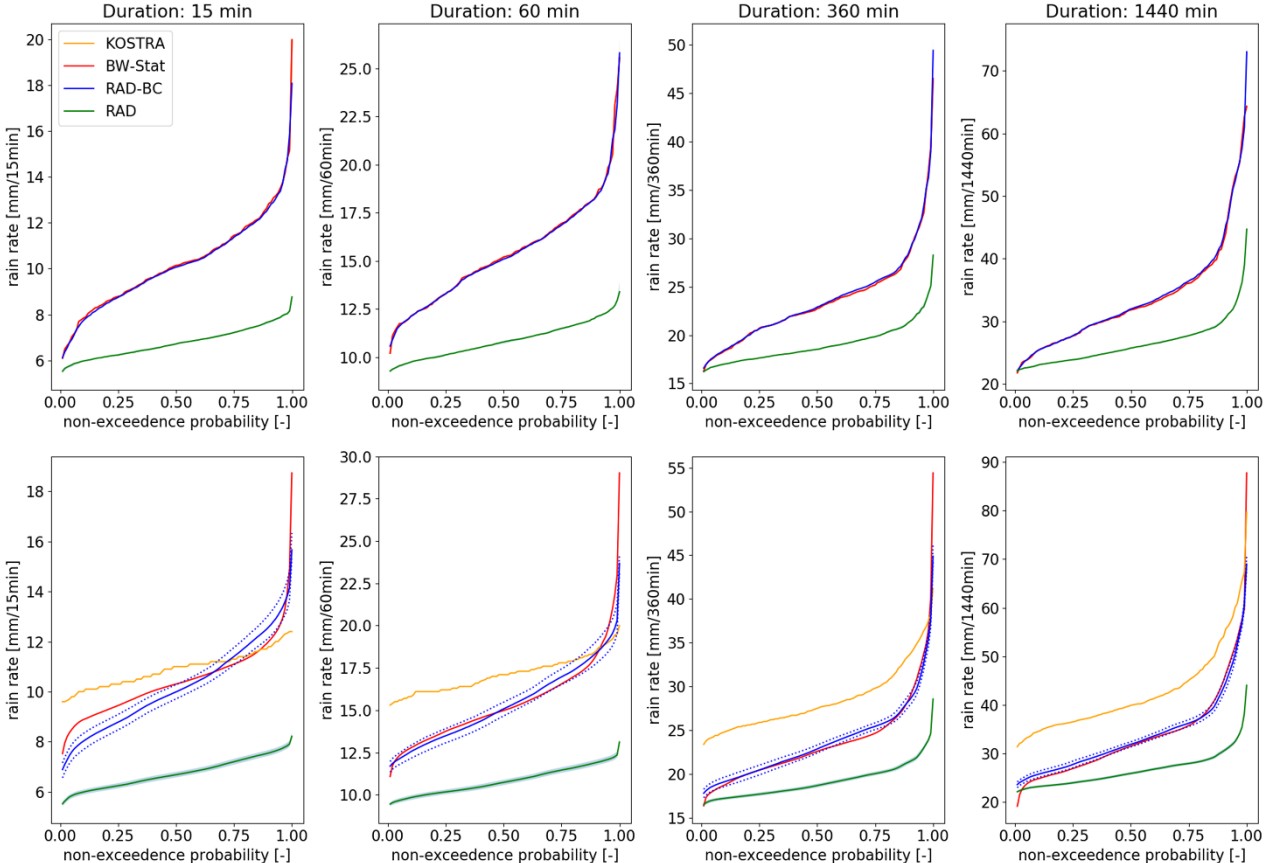

**Figure 3: Cumulative frequency distributions (CFD) of the location parameter for four different event durations when comparing**
**stations and radar data at the location of stations only (upper row) and integrated over the whole of BaWu (bottom row).) The dotted**
**blue lines in the bottom row represent the range of the five ensemble members (sampling uncertainty only, no bootstrapping).**



**Figure 4: Magnitude of design storms with a return rate of 100 years for four different event durations (15, 60, 360 and 1440 minutes,**
**depicted in rows) and three different datasets (KOSTRA, BW-Stat, RAD-BC, depicted in columns). Additionally, the difference**
**between BW-Stat and RAD-BC is depicted (right column).**





**Figure 5: Cumulative frequency distributions (CFD) of the magnitude of 20yr (upper row) and 100yr (bottom row) design storms for four different event and different datasets. The shaded range depicts the ensemble uncertainty (10th and 90th percentile of the range from the 1000 bootstraps for each of the 5 ensemble members). The dotted blue lines in the bottom row represent the range of the five ensemble members (sampling uncertainty, no bootstrapping) only. Note that in the BW-Stat dataset all values below/above the 5th/96th percentiles are set to the respective percentile value.**



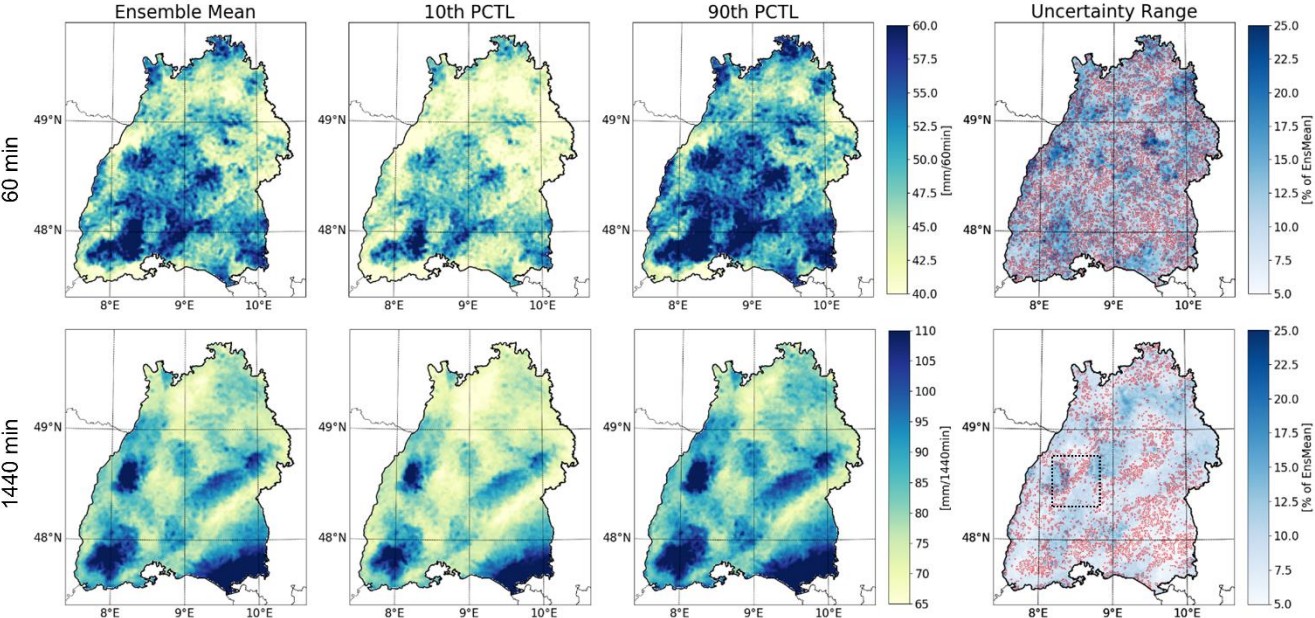


**Figure 6: Ensemble mean (left column) and the 10th and 90th percentiles (two middle columns) of a 100year design storm based for two durations (60 minute events – upper row; 1440 minute events – bottom row). Additionally, the ensemble uncertainty range (difference between the 90th and the 10th percentile of the full (bootstrapping & sampling) 5000 member) is depicted (right column). Regions, with a large (> 75% of the range) contribution of the sampling uncertainty are marked with red. The black dashed square in the panel in the lower right defines the northern Black Forest region discussed in the text.**




**Figure 7: Cumulative frequency distributions (CFD) of the scale (upper row) and shape (bottom row) parameter for the BW-Stat and RAD-BC datasets, when comparing stations and radar data at the location of all stations (left column) and for three different subsets filtered by the altitude of the respective station locations.**