# Peer review of "Enhancing the usability of weather radar data for the statistical analysis of extreme precipitation events"

_Hydrology and Earth System Sciences, 2021_

## Author Comment (AC1)

Comment on hess-2021-366
Francesco Marra (Referee)

Dear Francesco Marra,
thank you very much for your valuable critics, remarks and suggestions in order to improve our manuscript. Please find our response below to the various points you raised indicating what we adapted in the manuscript.
Additionally, I would like to apologize for not being active in the open discussion during the review phase. The reason for this is that I was on parental leave, which actually started a bit earlier than was originally foreseen.

Best regards, also on behalf of my co-author,
Andreas Hänsler

**Major comments:**
- The reference dataset is sometimes used to support the goodness of the new dataset and sometimes regarded as less accurate (e.g. in the patterns of sub-daily precip – see lines 16-18 in the abstract). Although the reasons behind this can be somehow understood, this is a problematic issue. On what bases is the dataset trusted as a reference (perhaps some durations are and some are not, some return periods are and some are not)?
I think a proper evaluation should rely on a trusted dataset. For example, rain gauges could provide a quantitatively trusted reference to gather information on the quantitative accuracy of the method on some selected locations. This might allow us to understand what aspects the radar product is or isn't able to reproduce (orographic influence at different durations, different return levels, etc). Alternatively, the trusted parts of the available dataset should be defined a priori and used for the validation, while the parts which are not trusted should be only used for comparison and discussion.

→ This is indeed a serious point but unfortunately not so easy to solve. We know that both reference datasets have serious shortcomings. Especially the spatial patterns in both datasets are impacted by the assumptions (e.g. orographic influence) of the interpolation routine. Furthermore, in the BW-Stats the magnitude of the design storms over mountainous regions can be influenced by the fact that often the same (although sometimes rather far apart) stations are pooled together. Hence, there is not the perfect reference dataset available, and especially the improvement of the spatial patterns was one of the main motivation aspects for our study.
But we agree that the shortcomings of the two reference datasets were not well described in the initial version of the manuscript. Hence we added much more information on this topic in the various parts (e.g. in the dataset description in Section 2.2. as well as in the discussion) of the manuscript.

- While I understand the need to avoid winter periods due to the known issues of weather radar monitoring with solid precipitation, it is not clear to me how it is possible to compare return levels derived from summer only (Apr-Oct as in this paper) with return levels derived from stations (the reference products) for durations up to 24 hours. The authors mention this at lines 82-84 ("Since we are mainly interested in short to medium range storm events that are mainly of convective nature, we only use data for the (summer) months from April to October, representing the main season for these kind of storm events"), but then durations up to 24 hours (e.g. see lines 244-255) are examined and discussed. This mismatch, which is not discussed by the authors, could also contribute to the overall bias found by the authors. I fear this might represent an important drawback of the presented product and of the presented comparison.

→ Actually the analysis design of the RADKLIM data is chosen in order to represent what was done when establishing the BW-Stat data. But this was unfortunately poorly described in the

original version of the manuscript and is now changed (e.g. see Section 2.2. on dataset description but also added this information in the description of the EVA - see Section 2.4). So also in BW-Stat only April to October data is used and the method was applied for 5 minutes to 24h.

With respect to convective vs. stratiform events it has to be mentioned that especially in April as well as during September/October frontal rains can lead to substantial rain amounts – hence including also the daily time step is important. Furthermore, since the main added value of the radar dataset is in the spatial picture we wanted to also analyze the shift in spatial patterns between short duration convective events and longer duration frontal rain events.

- I like the idea of sampling the surrounding pixels using probabilities, and I like the idea of basing the properties of the sampling pdf based on the typical size of convective rain cells in the region, but I am missing why the same mask is used for all durations. Since precipitation accumulations over longer durations are characterized by larger autocorrelation, my guess would be that 4 km might be good for short durations (even 1 hour could be border line according with what is said above), but too short for longer durations.

→ The reason for using the same mask for all durations is that we wanted to be able to refer differences in the spatial patterns between short and long durations to the data itself. If we would have changed the sample mask the resulting change in spatial patterns could be mainly due to this effect (or we would not know). With respect to autocorrelation we have a minimum temporal offset of 2 days required between analyzed events. Hence we assume that the effect of autocorrelation is rather small even if cells are pooled that are located not too far apart.

- Lines 107-111: this is presented in a confusing way. There is no guarantee that 100 years of data will provide perfect (or good for what matters) estimates of the 100-year return levels. Monte Carlo simulations run under realistic precipitation statistics show that empirical estimates will be subject to ~90% uncertainty (computed as the 90% confidence interval), while a simple GEV fit (method of the L-moments) will be subject to ~50% uncertainty. The advantage of using ~100 years of data instead of ~20 is clear, but should be presented in a better way.

→ That is certainly true. We changed it in the text.

- The results show an important systematic bias (as it can be inferred from fig. 4). This bias concerns most of the study area and cannot be seen as related to stochastic uncertainty, therefore the uncertainty quantification at section 3.4 cannot be accounted for explaining it. This is an important issue and I wonder what is the added value of such a quantitative information for the final user.

→ It simply shows the uncertainty related to the pooling and fitting and is in our opinion definitely something the user should be informed about, even if the absolute magnitude might not be directly in line with the one of the reference datasets.

To my view, this issue is related to a sub-optimal choice of the bias correction method (see details below), and addressing it should therefore be part of this study. The bias correction described in section 2.5 seems to me insufficient. Basically, this correction includes an additive adjustment to the data (changes the location parameter of the GPD). Since radar errors are far from being only additive, the resulting product is necessarily biased. Eventually, the results presented in the paper confirm this: the underestimation increases with return period, meaning that the other parameters are wrongly represented by the product and therefore also need to be adjusted. While the

authors mention these efforts as future directions, I think that the here presented results are not sufficient to justify this publication and that these additional efforts have to be invested here.

→ This is definitely an important point and we agree with the reviewer that the applied bias correction method is far away from being perfect. And of course we know from literature that especially for high intensity events the underestimation of precipitation in the radar data is enhanced,

although a recent study comparing RADKLIM data to station data points out that the rainfall magnitude of heavy rain events between radar and station is rather similar, but the frequency is not (Kreklow et al., 2020 – see manuscript for full reference). So this actually supports our conclusion that bias correction is a very complex issue and needs to be studied in more detail.

A possibility to overcome the fact that higher intensities seem to be affected by larger biases could of course be, to apply a multiplicative correction factor. But this would mean that we add rather large rainfall amounts to the highest intensities and also disrupt the homogeneity of the sampled events. So we decided against this approach.

Furthermore, as mentioned previously, the high magnitudes of design rainfalls in the BW-Stat dataset over mountainous regions can also be questioned and could also be an artefact of the pooling. And if we compare our dataset to KOSTRA for the short durations (e.g. 15 minutes or 1h) that can be attributed to have occurred mainly during the summer season, it is not necessarily true that the bias increases with the return period (see Figure 5 – left 4 panels).

So in short and as also mentioned in the manuscript we think that the correction is important and in the present form not absolutely satisfying. However, it is definitely a complex issue which needs more research and cannot be the focus of this study. We added some of the points discussed above to the discussion section of the manuscript to make it a bit clearer. But we still think that improving the bias correction method is beyond the scope of this study.

**Moderate comments:**

- It seems to me that larger ensembles could produce more accurate estimates (for example they could reduce the stochastic noise still present in the data and which required the smoothing of the maps). Why is a factor of 5 chosen? Are there only statistical-independence limitations or is it also a matter of computational time?

→ Yes, it is mainly due to computational issues. But also over mountain regions the number of ensemble members is limited as more and more identical cells will be sampled. We added both points to the manuscript.

- Lines 40-41: this is an over-simplification. The short record length is indeed among the important drawbacks of weather radar archives, but other issues were highlighted in literature. The most important one is definitely estimation inaccuracy: large systematic over- and under- estimations were found due to measurement errors (e.g. Eldardiry et al., 2015; Haberlandt and Berndt, 2016, among others), but in a recent review on the topic we also highlighted the inadequacy of the adopted statistical methods (Marra et al., 2019). As these aspects are somehow addressed by the methodology in this paper, I think the introduction should better present them.

→ We changed this in the introduction

- Section 2.2: information on the extreme value methodology used in the reference products has to be provided. Something is said later in the text, but the information should be presented in an organized manner here. Also, the implications of these choices should be discussed. For example, distributions with different tail heaviness will unavoidably show different biases at different return levels. If indeed different methodologies are used, the impact of these aspects on the comparison and on the results have to be discussed.

→There was definitely a lot of information missing with respect to the reference data. Hence we completely rewrote section 2.2

Lines 116-121: I am missing the relation between the typical size of the convective cells and sampling radius and normal distribution parameters.

→Over BaWu the size of a convective cell during summer is between 25 and 40 km. Hence we set the maximum radius to 25 km. The other parameters are chosen in a way that the sampled cells are relatively close to the COI.

Line 132: similar to the previous comment, why is 4 km chosen here?

→We analyzed that we do not sample additional events when we choosing the directly neighboring cells. That's why we set this minimum distance.

Line 220: It would be nice to see the results also for 1-year or 2-yr return levels. Since the adjustment is basically done on the 1-yr event, they should well isolate the quality of the product in relation to the bootstrap sampling method.

→Not sure if we fully understand this comment, but the respective figure actually shows a 1 yr event (with the difference that the loc parameter was not clipped below/above the 5th/95th percentile).

But as the second reviewer also pointed to the fact that it would be interesting to also see the spatial patterns of a 1yr event, we added a respective figure in the appendix and briefly discussed it in the text. Actually, the spatial patterns in the 1yr design storm of BW-Stat are similar to the ones of RAD-BC.

Line 232: why is the map smoothed? It seems this is to remove some noise. However, the noise we would see in these maps is a direct representation of the stochastic uncertainties affecting the overall methodology. I think the maps would be more informative without the smoothing.

→ Actually we applied the smoothing in order to make the maps a bit cleaner in order to not be distracted by very small scale features when comparing the RAD-BC data with the highly smoothed (due to underlying interpolation) BW-Stat dataset. But since we only applied a 3x3 cell smoothing, the small features are still very present in the final figure. See below the comparison between original (left) and non-smoothed (right) for two event durations (15 & 360 mins). Based on the small differences we think that we can keep the smoothed maps. Actually a significant portion of smoothing results from the applied ensemble approach.

[Figure]

Line 310-313: I might agree on the fact that higher-order moments are more difficult to estimate and to rely on, especially from "indirect" datasets such as the ones used here as a reference. I however,

think that this problem can be somehow addressed by using a more trusted reference and by using corresponding statistical methods.

→ To our knowledge there are not many other reference datasets available for BaWu than KOSTRA and the BW-Stat data. The latter one was actually compiled due to the known shortcomings of the KOSTRA dataset. So we agree that we have to work on this issue (see also comment above to the bias correction) but we think that this is a study itself.

Although not a native speaker myself, I felt that the language level could be improved, in part due to missing use or misuse of technical terms.

→Reviewer 2 suggested quite some changes regarding wording/language which we followed.

**Minor comments:**
Lines 16-18: this sentence is not completely clear. I could understand it only after reading the paper. Since this is the abstract, I suggest rewording it.

→Yes, indeed it is a bit misleading when you do not know the study itself. Hence, we adapted this sentence.

Line 32: some change-permitting GEV methods allow for changes also of the scale parameter (e.g. see Prosdocimi and Kjeldsen, 2021)

→Yes, that is true. But for the traditional method chosen in our study, our statement is true.

Lines 41-42: I personally disagree on this point. While this is very true for traditional methods based on extreme value analysis, there are some novel statistical methods which show promising results in this sense. They are now published since few years (the first papers are by Marani and Ignaccolo, 2015; Zorzetto et al., 2016), and many came after providing evidence (with applications to rain gauge data as well as satellite data) of the fact that 20 years might be sufficient for at-site estimates of even 100-year return levels. I believe it is time to recognize this by specifying that this limit concerns the traditional methods based extreme value analyses.

→ Thanks for pointing this out. I carefully had a look at the paper from Zorzetto et al (now also referenced in the revised version of the manuscript). While they definitely show that their MEV outperforms the more traditional methods when only short periods are considered it also becomes clear that the uncertainty/error of design storm estimates based on relatively short time periods are still larger than when analyzing longer records. Based on this, we believe that our original statement is not wrong, but we anyway changed the text in order to also point to these studies.

Line 112: it is not clear to me what the authors mean with "with underlying sampling probabilities"

→ Changed it to „with a specific sampling probability for each cell assigned "

Line 115: what does "not necessarily present" mean exactly? Is it a way to say "independent"?

→ It just means that we want to sample additional events or intensities which are not present in the COI – removed the word necessarily in the text. Independence of the events is assured by the 48 hour gap between two events included in the EVA.

Line 127: I suggest to include this information on the elevation range earlier in the text. Perhaps a short section describing the study area could help also in the following discussion.

→ We added the numbers in the introduction when we mention that the topography of BaWu is quite complex

Line 222: with "lower time steps", do you mean "shorter durations"?
→ Yes, exactly – we changed it accordingly

---

## Author Comment (AC2)

Comment on hess-2021-366
Anonymous Referee #2

Dear Anonymous Referee,
thank you very much for your valuable critics, remarks and suggestions in order to improve our manuscript. Please find our response to the various points you raised indicating what we adapted in the manuscript below.
Additionally, I would like to apologize for not being active in the open discussion during the review phase. The reason for this is that I was on parental leave, which actually started a bit earlier than was originally foreseen.

Best regards, also on behalf of my co-author,
Andreas Hänsler

**Major comments:**
1. A major concern is the minimum distance of the radar cells that are considered to statistically extend the time series of the cell of interest. As far as I understand the cells have to be at least 4 km apart. The authors mention that the typical size of a convective cell in Germany is 40 km for hourly events according to Lengfeld et al. 2019 (p.4, l.121 in this manuscript). Therefore, the minimum distance of 4 km seems a bit too small to me, especially when considering also daily events that have a much larger typical spatial extent. Did the authors perform any kind of independence check for the time series from the cells that are combined to a long time series, e.g. the correlation of the time series or the percentage of time steps with rainfall in the cell of interest but no rainfall in the other cells of the sample?
→ This is true that we sample rather close to the COI, in order to mainly sample cells that have similar rainfall characteristics. As shown in Figure 2b, the majority of sampled cells are in a distance range between 8 and 12 km. The events of the sampled cells will definitely have a certain amount of correlation (actually it is intended that they have) to the events in the COI - especially for the longer durations. However, since we have as prerequisite that single events (independent of the cell they are sampled from) have to be at least two days apart we assume that we can ignore the autocorrelation effect in the EVA, as the duration of an event is much shorter and hence the sample is independent of an event.

How do you make sure that the 258 events are actually taken from all 5 time series and not only taken from the 19 year time series of the COI?
→ Originally we did not investigate how often an event is pooled out of a specific cell, but we now looked at this. Indeed, we find, that each cell contributes almost the same amounts of events to the EVA (Median is 20%, 5th percentile is 13%, 95th percentile is 28%). We added this finding under section 3.1 in the manuscript.

I was also wondering if the same set of cells are used throughout the study or if the samples vary for the three durations that are considered.

→ The reason for using the same mask for all durations is that we wanted to be able to refer differences in the spatial patterns between short and long durations to the data itself. If we would have a change in the sample mask the change in spatial patterns could be mainly due to this effect.

2. The authors only consider precipitation data from April to October, because this is the main season of convective events of short durations. The statistical approach to determine designs storms

is based on a partial time series consisting of e (Euler's number) times the number of years. I was wondering, if this approach is still valid if only 7 out of 12 months of the year are considered.

→Yes, to our knowledge it is still valid. There are many studies available that use the Peak-over-Threshold (POT) approach for seasonal extreme value analysis (e.g. seasonal flood frequency analysis).

Although it is common knowledge that most of the convective storms occur during summer, some events might still be missed, especially for the design storms with 24 h durations that might also be associated with advective weather situations.

→ Yes, we agree that for the 24h case we will miss events that occurred outside the April to October season. We mainly included the 24h case since we wanted to demonstrate that there is actually a change in the spatial pattern between short duration and longer duration design storms when basing the analysis on spatially explicit data. This change in pattern is expected but not (or only to a lesser extend) visible in the interpolated station based reference datasets.

To my understanding, the reference data sets KOSTRA and BW-Stat consider all months and might not be comparable to the radar based data set. I would suggest to take all months into account or the authors should provide some kind of validation for their choice of selecting only summer months.

→ Actually the analysis design of the RADKLIM data is chosen in order to represent what was done when establishing the BW-Stat data. But this was unfortunately poorly described in the original version of the manuscript and is now changed (e.g. see Section 2.2. on dataset description but also added this information in the description of the EVA - see Section 2.4). So also in BW-Stat only April to October data is used and the method was applied for 5 minutes to 24h.

3. Section 2.2 about the reference data sets is quite short. More information about both data sets (e.g. how many stations are considered, length of the time series, interpolation methods, etc.) and on the differences in the statistical approaches to determine design storms from those data sets are desirable. The method for BW-Stat is briefly described in section 2.4. Maybe it would be better to have a general section about the methods first and then describe the data sets and their differences. E.g. that a two parameter GEV distribution is used in KOSTRA, instead of GPD for BW-Stats and the radar-based data set, is only mentioned in the discussion. This is important information that should be given in the method section.

→ We completely agree with the reviewer and completely changed the section on the reference datasets.

4. A more detailed description of the sampling process, the generation of the ensemble members, the bootstrapping method and the bias correction is needed to allow for better understanding of the results and of the choices made by the authors (e.g. why 5 ensemble members?).

→ The second reviewer also pointed towards some shortcomings in this part of the manuscript. So we added some more information in order to clarify his remarks, and hope that this now becomes clearer.

**Minor comments: -**
p.3, l.75: "... which leads to a spatially..." → "...which leads to spatially..."

→ Reviewer is right – we changed it in the manuscript

p.3, l.92: To my knowledge, the KOSTRA-DWD-2010R data set has a resolution of about 8.2 x 8.2 km. Did the authors perform some kind of remapping to achieve the 5 km x 5 km resolution?

→ Reviewer is right – we changed it in the manuscript

p.5, l151-152: Almost the same sentence is repeated on p.6, l.161-162.

→ Not exactly sure which sentence is repeated. On page 5 we talk about estimating the threshold value for the partial series, on page 6 we talk about parameter fitting for the GPD.

p.6, l.174-176: The radar data are adjusted to the 1 year design storms of the station-based BW-Stat data set. In the results section both data sets are also compared to design storms with 100 year return period derived from KOSTRA. For a better assessment of the differences between 100 year design storms from KOSTRA and the other two data sets it would be beneficial to also compare the 1 year design storms. Do they show the same features in the spatial pattern? How large are the differences?
→ We added a figure showing the spatial pattern of a 1yr event in the appendix and briefly discussed it in the text. Actually, the spatial patterns in the 1yr design storm of BW-Stat are similar to the ones of RAD-BC.

p7., l.205: "... located more the centre..." → "...located more to the centre..."
→ Reviewer is right – we changed it in the manuscript

p.8, l.222: "...time steps can attributed..." → "...time steps can be attributed..."
→ Reviewer is right – we changed it in the manuscript
What is meant by "lower spatial distribution"? Lower spatial resolution?
→ Yes, we wanted to talk about the spatial resolution. We changed it in the manuscript

p.8, l.230: "...as well as the for the bias-corrected..." → "...as well as for the bias-corrected..."
→ Reviewer is right – we changed it in the manuscript

p.8, l.238: Isn't the 24 h design storm from KOSTRA shown in Fig.4? Or is that something else the authors refer to here?
→ Yes, it is shown. But the 360 minute storm is actually interpolated between 1 and 12h and the latter is not shown. We changed it accordingly and actually also pointed to the fact, that in the latest version also the 24h design storm is interpolated (between 12h and 72h).

p.8, l.249: REGNIE is first mentioned here and should be explained.
→ REGNIE is now introduced earlier in section 2.2

p.9, l.257: Which figure do the author refer to here regarding the 20 year design storms?
→ Figure 5, like the rest of the paragraph

p.9, l.269: 10 th → 10th
→ Reviewer is right – we changed it in the manuscript

p.9, l.276: "... in in the case..." → "...in the case..."
→ Reviewer is right – we changed it in the manuscript

p.9, l.283: "...relatively larger uncertainty..." → "...relatively large uncertainty..." or "...larger uncertainty..."
→ Reviewer is right – we changed it in the manuscript to „...relatively large uncertainty..."

p.10, l.299-303: This might fit better in the result section.
→ We actually would like to leave it in the discussion section since it fits to the overall discussion of the remaining biases

p.10, l.310: "...can be seen as a rather robust..." → "...can be seen as rather robust..."

→ Reviewer is right – we changed it in the manuscript

p.11, l.318: "...difference...become..." → "...difference...becomes..."
→ we wanted to talk about differences – not a difference – so we changed it to „differences … become ..."

p.11, l.319: "...between in rainfall estimates..." → "...between rainfall estimates..
→ Reviewer is right – we changed it in the manuscript

p.11, l.331: "...in future" → "...in the future"
→ Reviewer is right – we changed it in the manuscript

p.14, l.417-418: "...based on distance to cell of interest..." → "...based on the distance to the cell of interest..."
→ Reviewer is right – we changed it in the manuscript

p.14., l.420: "...is marked with in red..." → "...is marked in red..."
→ Reviewer is right – we changed it in the manuscript
April to October of which years?
→ The most recent 30 year period from 1991 to 2020 – we added this information to the figure caption

p.15, l.426: It should either be "for a single member" or "for single members"
→ Reviewer is right – we changed it in the manuscript to for a single member.

p.15, l.430: Remove one of the brackets.
→ Done

p.16, Figure 4: It would also be interesting to see the differences between KOSTRA and RAD-BC.
→ we decided not to plot the 1 to 1 differences as the methodology and the spatial resolution is very different to BW-Stat and RAD-BC. We included it rather as an independent reference. We hope that this becomes clearer now, since we substantially changed the description of the datasets.

p.17, l.440: What is meant by "four different event"? I assume it is supposed to be "for different event durations"?
→ Reviewer is right – we changed it in the manuscript

p.17, l.443: Why is the 96th percentile chosen here instead of the 95th percentile?
→ it is the 95th percentile - we changed it.

P18, l.450: There is no comma needed after "regions".
→ Reviewer is right – we changed it in the manuscript